# Oncological Outcomes of Patients with High-Volume mCRPC: Results from a Longitudinal Real-Life Multicenter Cohort

**DOI:** 10.3390/cancers15194809

**Published:** 2023-09-29

**Authors:** Mariaconsiglia Ferriero, Francesco Prata, Umberto Anceschi, Serena Astore, Alfredo Maria Bove, Aldo Brassetti, Fabio Calabrò, Silvia Chiellino, Cosimo De Nunzio, Gaetano Facchini, Elisena Franzese, Michela Izzo, Riccardo Mastroianni, Leonardo Misuraca, Richard Naspro, Rocco Papalia, Annalisa Pappalardo, Giorgia Tema, Gabriele Tuderti, Beatrice Turchi, Andrea Tubaro, Giuseppe Simone

**Affiliations:** 1Department of Urology, IRCCS “Regina Elena” National Cancer Institute, 00144 Rome, Italy; umberto.anceschi@ifo.it (U.A.); alfredo.bove@ifo.it (A.M.B.); aldo.brassetti@ifo.it (A.B.); riccardo.mastroianni@ifo.it (R.M.); leonardo.misuraca@ifo.it (L.M.); gabriele.tuderti@ifo.it (G.T.); giuseppe.simone@ifo.it (G.S.); 2Department of Urology, Fondazione Policlinico Universitario Campus Bio-Medico, 00128 Rome, Italy; f.prata@unicampus.it (F.P.); rocco.papalia@policlinicocampus.it (R.P.); 3Department of Medical Oncology, San Camillo-Forlanini Hospital, 00149 Rome, Italy; serena.astore@scamilloforlanini.rm.it; 4Department of Oncology, IRCCS Fondazione Policlinico San Matteo, 27100 Pavia, Italy; s.chiellino@smatteo.pv.it (S.C.); 5Department of Urology, Faculty of Health Sciences, “Sapienza” University, Ospedale Sant’Andrea, 00185 Rome, Italy; cosimodenunzio@virgilio.it (C.D.N.); beatrice.turchi@uniroma1.it (B.T.); andrea.tubaro@uniroma1.it (A.T.); 6Medical Oncology Complex Unit, “Santa Maria della Grazie” Hospital, ASL Napoli 2 Nord, 80078 Pozzuoli, Italy; 7Department of Urology, IRCCS Fondazione Policlinico San Matteo, 27100 Pavia, Italy; r.naspro@smatteo.pv.it (R.N.)

**Keywords:** prostate cancer, mCRPC, chemotherapy, ARTA, high volume

## Abstract

**Simple Summary:**

Metastatic prostate cancer (PCa) may occur as progression after local therapy with curative intent or may be metastatic as newly onset. Approximately 5% of PCa present with metastatic stage at the time of diagnosis. Therapeutic management differs widely according to the site of metastases, the sensitivity to hormonal treatments and the volume of disease. We reported survival outcomes of patients with high-volume metastatic castration-resistant PCa (mCRPC) treated with ARTA in a chemo-naïve setting compared to patients treated with chemotherapy as first-line from a longitudinal real-life multicenter series. In our population of 88 high-volume-disease mCRPC patients, we showed that survival probabilities are comparable between first-line ARTA and upfront chemotherapy-treated cohorts. Therefore, regardless tumor burden, novel antiandrogens can be useful treatment options and could be considered as first-line in order to postpone the use of more toxic treatments such as chemotherapy, in the case of significant disease progression.

**Abstract:**

Registrative trials recommended the use of upfront chemotherapy in high-volume metastatic prostate cancer. We reported survival outcomes of patients with high-volume mCRPC treated with ARTA in a chemo-naïve setting compared to patients treated with chemotherapy as first-line from a longitudinal real-life multicenter series. We retrospectively collected data on mCRPC patients treated at six centers. The dataset was queried for high-volume disease (defined as more than 6 bone lesions or bulky nodes ≥ 5 cm). We compared the main clinical features of chemo-naïve versus chemo-treated patients. The Mann–Whitney U test and Chi-squared test were used to compare continuous and categorial variables, respectively. The Kaplan–Meier method was used to compare differences in terms of progression-free survival (PFS), cancer specific survival (CSS) and overall survival (OS) in an upfront ARTA or chemo-treated setting. Survival probabilities were computed at 12, 24, 48, and 60 months. Out of 216 patients, 88 cases with high-volume disease were selected. Sixty-nine patients (78.4%) received upfront ARTA, while 19 patients received chemotherapy as the first-line treatment option. Forty-eight patients received Abiraterone (AA), 21 patients received Enzalutamide (EZ) as the first-line treatment. The ARTA population was older (*p* = 0.007) and less likely to receive further lines of treatment (*p* = 0.001) than the chemo-treated cohort. The five-year PFS, CSS and OS were 60%, 73.3%, and 72.9%, respectively. Overall, 28 patients (31.8%) shifted after their first-line therapy to a second-line therapy: EZ was prescribed in 17 cases, AA in seven cases and radiometabolic therapy in four patients. Sixteen cases (18.2%) developed significant progression and were treated with chemotherapy. At Kaplan–Meyer analysis PFS, CSS and OS were comparable for upfront ARTA vs chemo-treated patients (log rank *p* = 0.10, *p* = 0.64 and *p* = 0.36, respectively). We reported comparable survival probabilities in a real-life series of high-volume mCRPC patients who either received upfront ARTA or chemotherapy. Patients primarily treated with chemotherapy were younger and more likely to receive further treatment lines than the upfront ARTA cohort. Our data support the use of novel antiandrogens as first line treatment regardless tumor burden, delaying the beginning of a more toxic chemotherapy in case of significant disease progression.

## 1. Introduction

Prostate cancer (PCa) ranks as the second most common cancer in men worldwide, after lung cancer. In Italy, it takes the lead as the most diagnosed neoplasm, with approximately 5% of patients presenting with metastatic disease at the time of diagnosis [1]. Advanced PCa can arise from a relapse following local curative-intent therapy like radical prostatectomy or radiation therapy, or it may present as metastatic from the onset. Metastatic PCa management differs widely according to the stage, the sensitivity to hormonal therapy and the volume of disease. Until recently, androgen deprivation therapy (ADT) was considered the standard of care for metastatic PCa. However, the introduction of several novel androgen receptor targeted agents (ARTAs) significantly changed the natural history of the disease. Consequently, urologists, as primarily surgeons, now play an active role in the management of metastatic disease. They are integral members of multidisciplinary boards, participating in decision-making alongside oncologists and radiation therapists. According to European Association of Urology (EAU) guidelines, castration-resistant prostate cancer (CRPC) is defined by the biochemical or radiological progression of disease combined with castrate serum testosterone level (<50 ng/dL or 1.7 nmol/L) [2]. Docetaxel, an anti-mitotic chemotherapy agent, was the initial life-prolonging therapy approved for the treatment of metastatic castration-resistant prostate cancer (mCRPC) [3,4]. Recently, chemotherapy for metastatic disease has been reserved for cases with a high tumor burden, either in metastatic hormone sensitive prostate cancer (mHSPC) or in an mCRPC setting after failure of one or more ARTA lines of therapy, due to a less tolerable toxicity rate. The development of resistance to the main primary treatment regimens remained an unsolved issue [4]. The optimal management of PCa, given its long natural history, involves carefully selecting treatment options that provide the longest sensitivity to the current drug. Common ARTA, such as Abiraterone Acetate (AA) or Enzalutamide (EZ), are widely used. However, both lines of therapy may experience a decline in effectiveness over time with the progression of the disease. A crucial factor in stratifying the management of mCRPC is the tumor burden (high- versus low-volume), which comes with different available definitions. CHAARTED and GETUG-AFU 15 used the same definition for high-volume disease, such as the presence of visceral metastases and/or at least four bone lesions, with at least one outside of the vertebral column and/or pelvis [5,6]. Moreover, the STAMPEDE trial explored the efficacy of radiation therapy vs standard of care in newly diagnosed metastatic PCa patients [7], in which high-volume metastatic disease was defined according to CHAARTED trial. According to subgroup analysis, there was no benefit in any treatment options in patients with high metastatic tumor burden (HR 1.07, 95% CI 0.90–1.28; *p* = 0.42). It follows that no conclusive recommendations can be made. [7] International guidelines do not establish a specific sequence or preferred treatment for chemo-naïve mCRPC patients but list all treatment options in alphabetical order and recommend discussing the case in a multidisciplinary team. Factors such as previous treatments, symptoms, co-morbidities, genomic profile, extent of disease and patient preference should be considered in decision-making processes [2].

However, for patients with high-volume metastatic disease, upfront chemotherapy remains an indicated approach [8].

Consequently, the challenge within the context of the mCRPC setting lies precisely in the absence of disease volume definitions, dedicated studies and established recommendations concerning the most effective treatment sequencing, especially in those presenting with high-volume disease.

To overcome this issue, we previously defined high-volume disease based on the radiological extent, as bulky positive nodes (≥5 cm) or more than six bone metastases at mCRPC onset and showed that high-volume was a significant predictor of progression-free survival probability (PFS) [9].

The use of chemotherapy upfront often comes with a significant rate of toxicity and its application is rooted in studies conducted prior to the era of ARTA. As a result, a unanimous consensus recommendation that outlines the optimal sequencing of systemic therapeutic agents remains a need. On this background, we reported survival outcomes of high-volume mCRPC patients treated with upfront ARTA in a chemo-naïve setting compared to patients treated with chemotherapy as first-line in a longitudinal real-life multicenter series.

## 2. Materials and Methods

### 2.1. Patient Population

Between July 2014 and June 2022, we retrospectively collected real-life data on mCRPC patients treated at six centers, who either received AA or EZ or Docetaxel as first- or second-line treatment. Patients treated with ARTA or chemotherapy in a castration sensitive setting were excluded.

All patients had castrate levels of testosterone (<50 ng/dL), with ongoing ADT and increasing PSA levels.

Metastatic disease was defined as the occurrence of bone, visceral and/or lymph node metastases at any site other than pelvic nodes.

The dataset was queried for “high-volume disease”.

High tumor burden was defined based on the radiological extent of disease, as bulky positive nodes (≥5 cm) or more than 6 bone metastases at mCRPC [9]. Symptoms were not considered a reason for defining a patient as high-volume.

### 2.2. Treatment Regimens

All mCRPC patients received 1000 mg of AA plus 10 mg of Prednisone or 160 mg of EZ daily or Docetaxel as first- or second-line therapy. The selection of treatment as upfront regimen was arbitrarily made by oncologists or urologists according to local protocol and disparate criteria (age, comorbidities, tolerance, patient compliance, drugs availability, physician’s preference). Patients experiencing significant toxicity or disease progression were shifted to a second line of therapy.

### 2.3. Staging and Follow-Up

All patients were staged according to EAU guidelines with a computed tomography (CT) scan and bone scan for disease progression under ADT. According to physician discretion, positron emission tomography (PET)/CT scan was performed when conventional imaging was negative and there was a clinical suspicion of progression of the disease. Prostate-specific membrane antigen (PSMA) was used as the preferred radio tracer of the PET/CT scan when available. Physical examination as well as laboratory routine biochemistry were carried out at baseline and subsequently at 3-month intervals. Patients were visited monthly and imaging re-evaluation was performed every 6 months regardless of the PSA serum levels and symptoms. Clinical features, treatment outcomes and toxicity were recorded at each visit [9].

### 2.4. Outcomes

The main outcomes included PFS, cancer-specific survival (CSS), and overall survival (OS) probabilities for both regimens. PFS probability was defined as time from the first dose of upfront treatment to the first radiographic evidence of progression [10].

### 2.5. Statistical Analysis

We compared the clinical features of two cohorts, upfront ARTA versus chemo-treated patients. Continuous variables were presented as median and interquartile ranges (IQRs) and were compared using either the Mann–Whitney U test or a Kruskal–Wallis one-way analysis based on their normal or not-normal distribution, respectively (normality of the distribution of variables was tested by the Kolmogorov–Smirnov test). Frequencies and proportions were used to report categorical variables that were compared using Chi-squared test. A two-sided *p*-value < 0.05 was considered statistically significant. The PFS, CSS, and OS were assessed with the Kaplan–Meier method and the log rank test was applied to assess statistical significance between groups. Survival probabilities were computed at 12, 24, 48, and 60 months. The IBM Statistical Package for the Social Sciences (SPSS) statistical software package (IBM Corp. Released 2020. IBM SPSS Statistics, Version 27.0. IBM Corp., Armonk, NY, USA) was used for statistical analyses.

## 3. Results

Out of 216 patients, 88 cases with high-volume disease were selected. Sixty-nine patients (78.4%) were chemo-naïve and treated with upfront ARTA, while 19 patients received chemotherapy as their first treatment option. Forty-eight patients received Abiraterone (AA), 21 patients received Enzalutamide (EZ) as the first treatment line. Clinical features of the whole cohort are reported in Table 1.

The upfront ARTA population was older (*p* = 0.007) and less likely to receive further lines of treatment (*p* = 0.001) than the chemo-treated cohort (Table 2).

Five-year PFS, CSS, and OS were 60%, 73.3%, and 72.9%, respectively. Overall, 28 patients (31.8%) after a first-line shifted to a second-line therapy: EZ was prescribed in 17 cases, AA in seven cases, and radiometabolic therapy in four patients. Sixteen cases (18.2%) developed significant progression and were treated with chemotherapy. At Kaplan–Meyer analysis PFS, CSS, and OS were comparable between upfront ARTA vs chemo-treated patients (log rank *p* = 0.10, *p* = 0.64 and *p* = 0.36, respectively, Figure 1).

## 4. Discussion

In this study we performed a retrospective evaluation from a longitudinal real-life multicenter series of high-volume mCRPC patients treated with chemotherapy versus ARTA (either AA or EZ) as the first treatment line. According to a high-level of evidence, the EAU guidelines recommend considering the early application of chemotherapy, beginning with docetaxel and potentially progressing to cabazitaxel in the treatment sequence, if patients are able to tolerate those regimens [2]. Nevertheless, determining the most effective first-line therapy for CRPC remains challenging, as no validated predictive factors have been established. Considerations such as performance status, symptoms, co-morbidities, disease location and extent, genomic profile, patient preference, and prior treatments all play pivotal roles, making difficult to provide a definitive recommendation [5]. The available data are solely derived from real-life experiences, reporting heterogeneous information due to the treatment selection made at physician discretion. Consequently, there is a pressing need for prospective comparative studies.

A real-life multicentric series showed that patients receiving EZ as first-line treatment had significantly higher PSA response (95.9% vs. 67%, *p* < 0.001), comparable toxicity rate (10.2% vs. 16.3%, *p* = 0.437), and PFS probabilities (*p* = 0.145) compared to AA, while EZ or radiometabolic therapy as second-line treatment displayed equivalent toxicity and PSA response rates to those observed on first-line (11.1% vs. 12.4%, *p* = 0.77; 73.1% vs. 77.4%, *p* = 0.62, respectively) [9]. Moreover, 2-year PFS, CSS, and OS probabilities were comparable between first- and second-line cohorts (12.1% vs. 16.2%, *p* = 0.07; 85.7% vs. 86.4%, *p* = 0.98; 71% vs. 80.3%, *p* = 0.66, respectively) [9]. In a more recent series of 117 chemo-naïve mCRPC patients who received AA or EZ as first-line therapy, eight patients underwent salvage chemotherapy after first-line failure and 28 patients shifted to a second-line therapy [11]. A retrospective series showed a favorable PFS of the ARTA–Docetaxel sequence (HR 0.38; 95% CI 0.24–0.59; *p* < 0.001) than ARTA–ARTA, suggesting better oncological outcomes for chemotherapy as second-line treatment [12]. Similarly, an observational retrospective real-life study reported improved PSA response rate and longer time to PSA progression (adjusted OR = 2.27, *p* = 0.005; adjusted HR = 0.66, *p* = 0.010, respectively) with second-line chemotherapy in mCRPC experiencing early progression after AA or EZ [13].

Conversely, there is a recent increased interest in combining upfront chemotherapy with ADT + ARTA in high-volume patients in a castration sensitive setting. Two multicenter phase 3 randomized clinical trials, PEACE-1 and ARASENS, showed that the combination of ADT, docetaxel, and either AA or darolutamide provided a similar OS benefit [14,15]. Both trials demonstrated that the triplet therapy outperformed ADT-Docetaxel alone, suggesting ADT-Docetaxel-Darolutamide as the established standard of care for de novo mHSPC. However, the higher toxicity rates were primarily associated with the use of chemotherapy. Additionally, these trials did not investigate whether the triplet regimen provided comparable benefits in terms of both PFS and OS compared to the doublet ADT-AA or ADT-Darolutamide. Doublet therapy may find more common use in current medical practice owing to its superior safety profile and simplified administration. On the other hand, the triplet regimen should be reserved for highly selected cohorts of patients with high-volume disease.

Other studies investigated the oncological outcomes of initiating chemotherapy as the primary treatment in patients with metastatic PCa.

Miura Y. et al. reported that upfront intensive therapy with Docetaxel or AA did prolong CRPC-free survival (*p* = 0.022) after stratification for age ≥75 years [16]. On the contrary, Narita S. et al. showed how upfront AA provided better CRPC-free survival than upfront chemotherapy (*p* = 0.002); however, no significant differences in second-line PFS or OS were observed between the two groups (*p* = 0.254 and *p* = 0.444, respectively) [17]. In our study, we showed comparable survival outcomes for either upfront ARTA and chemo-treated cohorts in terms of PFS, CSS, and OS probabilities (log rank *p* = 0.10, *p* = 0.64 and *p* = 0.36, respectively). Hence, due to conflicting results, making definitive decisions regarding the optimal treatment is challenging. Patients with high tumor burden may respond differently to treatment regimens, with chemotherapy being the preferred first-line choice. However, there are currently no specific recommendations based on tumor burden. High-volume disease demonstrated its significance as a predictor of OS, but the therapeutic impact of EZ and AA on high-volume mCRPC remains a subject of debate [18]. Previous studies showed that high-volume mCRPC cohort had significantly lower PFS compared with low-volume mCRPC cohort (*p* = 0.015) [9]. Data from the PREVAIL study showed that chemo-naïve mCRPC patients treated with EZ with low- or high-volume disease, as well as a subgroup of 110 patients with low PSA levels (<10 ng/mL) and high tumor burden treated with EZ, displayed comparable radiological progression rates when compared to low-volume disease [13,19,20]. We investigated survival outcomes in the high-volume setting; our results suggested that alternative first-line regimens should be considered instead of upfront chemotherapy, especially in high-volume mCRPC cohorts. Notably, PFS, CSS, and OS were comparable in our real-life population regardless tumor burden. Moreover, patients treated with upfront chemotherapy were younger (*p* = 0.007) and more likely to receive further treatment lines than the upfront ARTA cohort (*p* = 0.001). These findings support the prevailing approach of using ARTA for mCRPC, a widely adopted choice in clinical practice. This is primarily due to the more favorable toxicity profile in contrast to upfront chemotherapy, disease progression or clinically significant toxicity being the main drivers for switching treatments. Docetaxel-related adverse events include infusion reactions, febrile neutropenia, fatigue, fluid retention, pneumonitis, cutaneous and nail toxicity, epiphora and lacrimal duct stenosis, gastrointestinal complications, and neuropathies. However, peripheral neuropathy is a long-term side effect of taxane chemotherapy that may be debilitating for patients well after completion of treatment [21]. On the other hand, long-term therapy with AA plus prednisone may potentially produce steroid-induced adverse events, such as hyperglycemia or diabetes, sarcopenia, hypertension and cardiovascular risk, osteoporosis, adrenal insufficiency, and infections, even when a low dose is administered [17]. Therefore, by identifying potential candidates for upfront ARTA in high-volume mCRPC population, we could minimize the chemotherapy overexposure. Hence, a meticulous approach to patient selection for upfront chemotherapy is imperative, by selecting potential candidates who may not derive a substantial survival benefit and might even face potential toxicity.

The use of local treatment for mCRPC has gained popularity, particularly in oligopregressive disease. Several studies have highlighted its efficacy in cases of low metastatic burden, particularly in terms of PFS. In our recent multicentric real-life experience, we observed that patients with low-volume disease displayed notably higher PFS probabilities (log rank *p* = 0.009) compared to those with high-volume disease. Furthermore, loco-regional treatments displayed a significant impact on PFS of low tumor burden patients (log rank *p* = 0.048). Nevertheless, a negligible impact has been proved in the whole cohort as among patients with high-volume disease (log rank *p* = 0.6 and *p* = 0.75) [11]. Another study including 300 metastatic PCa patients showed that locoregional treatment was more frequently performed in patients with low-volume disease (35.4% vs. 16.2%; *p* < 0.001), lower serum PSA at diagnosis (*p* = 0.005), and local symptoms (34.2% vs. 4.8%; *p* < 0.001), and was an independent predictor of longer OS at multivariable Cox analysis (62.1 vs. 55.8 months; HR 0.74; *p* = 0.044) [22]. Although the present study included 88 patients with high-volume disease who were treated with AA (*n* = 48) or EZ (*n* = 21) as first-line treatment, we could not address the real benefits of systemic treatment on prognosis because of the small sample size and many selection biases. Validation studies to assess the advantages of different systemic treatments in mCRPC patients are still needed.

Therefore, given the current lack of evidence for first-line treatment selection, shared decision-making through informed consent is crucial for the best treatment choice after discussion on potential benefits and disadvantages in each single case.

We acknowledge several limitations for this study. First of all, the retrospective nature and the relatively short follow-up, as well as the lack of a widespread consensus on the optimal treatment sequencing for mCRPC patients. The small, unbalanced sample size and the lack of control over potential confounding factors for treatment response may impact our results. Another potential limitation is the heterogeneity of the whole cohort, due to different ages, comorbidities, the inclusion of both metastatic de novo patients and those initially non-metastatic but progressed during the follow-up, and also physician’s preference for treatment choice between urologists and oncologists of different centers. Moreover, the use of PET/CT scan for the definition of metastatic disease, the choice of drug and the shift to another treatment line, as well as the lack of central radiologic review for clinical staging, have to be considered as other potential limitations. Larger and more homogeneous series are needed to answer the unsolved issues and to draw robust and definitive conclusions. We reported comparable survival outcomes regardless tumor volume and chemotherapy regimen, supporting the efficacy of upfront antiandrogens irrespective of tumor burden. Further studies are necessary to externally validate our results and understand how to select the mCRPC patient who can benefit from upfront ARTA instead of chemotherapy.

Nevertheless, our study showed a portrayal of a real-world scenario, thereby producing results that are more broadly generalizable than clinical trials. These results can be applied to a larger population of high-volume mCRPC patients, with different baseline characteristics and demographic profiles.

## 5. Conclusions

We reported comparable survival probabilities in a real-life series of high-volume mCRPC patients, regardless first line therapy. Patients previously treated with chemotherapy were younger and more likely to receive further treatment lines than upfront ARTA cohort. These data support the use of novel antiandrogens as first-line treatment regardless tumor burden, delaying the beginning of a more toxic chemotherapy in case of significant disease progression.

## Figures and Tables

**Figure 1 cancers-15-04809-f001:**
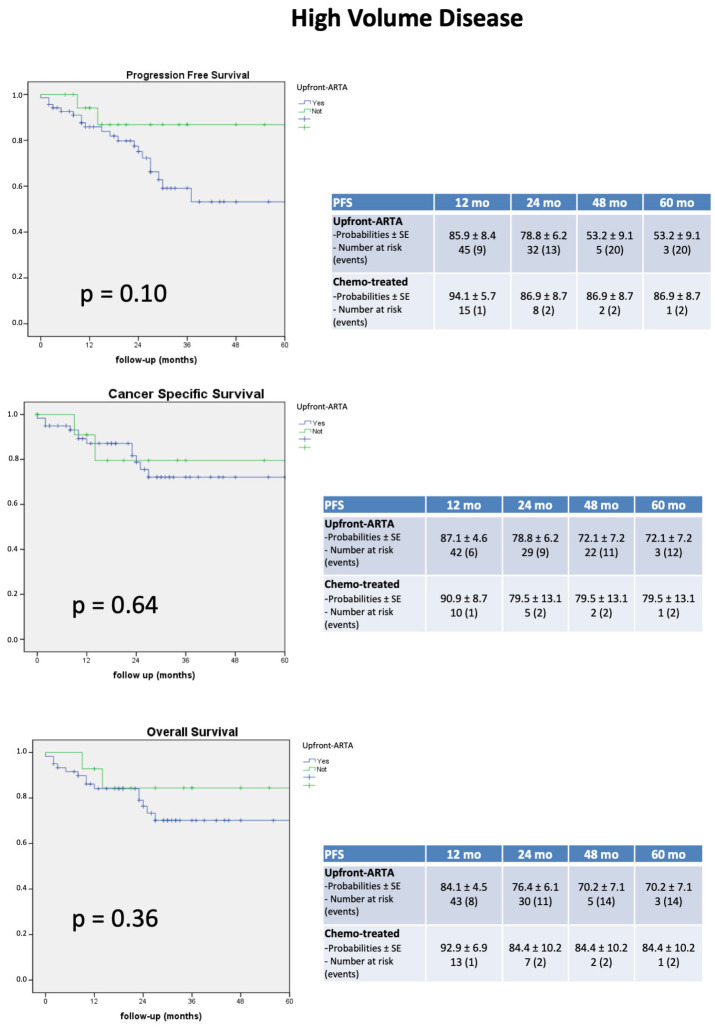
Kaplan–Meier curves showing progression free survival (PFS), cancer specific survival (CSS) and overall survival (OS) of upfront ARTA versus chemo-treated cohort.

**Table 1 cancers-15-04809-t001:** Clinical features of the whole cohort.

Clinical Features	Median or N (IQR or %)
**Age (years)**	74 (67–79)
**ECOG**	
0–1	85 (96.6)
2	3 (3.4)
**ISUP Grade Group (%)**	
NA	8 (9.1)
1–2	6 (6.8)
3	20 (22.7)
4	27 (30.7)
5	27 (30.7)
**Baseline Staging PCa (%)**	
**cT**	
T1–2	68 (77.3)
T3	16 (18.2)
T4	4 (4.5)
**cN 1**	20 (22.7)
**cM1**	13 (14.8)
**Local Treatment (%)**	
- Radical Prostatectomy	7 (8)
- Radiation Therapy	17 (19.3)
- None	45 (51.1)
- Both	19 (21.6)
**ADT Lenght (mo)**	18 (11–47)
**ADT Lines (N)**	1 (1–1)
**Time to CRPC (years)**	13 (7–22)
**PSA CRPC (ng/dL)**	15.1 (4.2–58.2)
**cN 1 CRPC (%)**	56 (63.6)
**Follow Up (mo)**	21 (10–30)
**Upfront ARTA (for chemo naïve cohort)**	69 (78.4)
Abiraterone Acetate	48 (54.5)
Enzalutamide	21 (45.5)
**Second Line Drug**	28 (31.8)
Abiraterone Acetate	7 (8)
Enzalutamide	17 (19.3)
Lutetium	1 (1.1)
Radium-223	3 (3.4)
**Target Therapy**	
SBRT	23 (26.1)
Lymph Node Dissection	0

**Table 2 cancers-15-04809-t002:** Clinical features of upfront ARTA versus chemo-treated cohort.

Feature	Upfront ARTA	Chemo-Treated	*p*
N (%) or Median (Range)	69	19	
**Age, years**	75 (69–81)	69 (63–74)	**0.007**
**ECOG**			
**0**	49 (71)	12 (63.1)	0.76
**1**	18 (26.1)	6 (31.6)	
**2**	2 (2.9)	1 (5.3)	
**cN1 (at diagnosis)**	48 (69.6)	8 (42.1)	0.84
**cM1 (at diagnosis)**	10 (14.5)	3 (15.8)	0.88
**PSA at CRPC (ng/mL)**	13.8 (4.3–46.9)	18 (3.5–376)	0.33
**Time to CRPC (months)**	13 (7–22)	13 (11–24)	0.45
**cN at CRPC**	48 (69.6)	8 (42.1)	0.034
**ADT Lines, N**			
**1**	53 (76.8)	17 (89.5)	0.23
**2**	16 (23.2)	2 (10.5)	
**CRPC Treatment Lines, N**			
**1**	42 (60.9)	2 (10.5)	**0.001**
**2**	16 (23.2)	9 (47.4)	
**3**	7 (10.1)	6 (31.6)	
**4**	4 (5.8)	2 (10.5)	
**Target Therapies**	17 (24.6)	6 (31.6)	0.54

## Data Availability

The data presented in this study are available if requested.

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
