# Peer review of "Oncological Outcomes of Patients with High-Volume mCRPC: Results from a Longitudinal Real-Life Multicenter Cohort"

_cancers, 2023, doi:10.3390/cancers15194809_

Round 1

Reviewer 1 Report

This is a multicentre retrospective study on mCRPC patients. Authors have focused their analysis on what they have characterized on previous publication as "High-volume disease"

In the specific article the effect of previous chemotherapy was analyzed. 

There are several problems in this article that should be improved:

1. Scientific hypothesis for this analysis as stated by the authors is "The lack of a general consensus recommendation for the optimal sequencing of systemic therapeutic agents"  and they have "reported survival outcomes of patients with mCRPC from a longitudinal real-life multicenter series" 

However, what they present is survival outcomes of what considered "High-Volume disease" in chemo-treated vs chemo-naive mCRPC patients. The aim of the analysis should be clearly stated in their introduction. 

2. What is the role of "High-Volume" disease in the mCRPC setting? There is a general confusion in the article betwen mCRPC and mHSPC setting. The High-volume disease definition was initially introduced in mHSPC setting and its role was confirmed as prognostic biomarker. It also allowed to differentiate treatment pathways between high- and low-volume patients. However, current definition for mCRPC disease lack generalized validation and was not used in consensus statements or guidelines for treatment selection. Other parameters eg. symptomatic disease that was earlier used was not included in the current analysis. 

3. In the time period analyzed, chemotherapy and ARTAs were also approved for the mHSPC setting and obviously such treatments affect treatment selection and prognosis in mCRPC. There is no information regarding the disease pathway for these patients and for those that were castration resistant after mHSPC treatment there is no information regarding treatment in the mHSPC setting. 

4. Treatment sequencing rather than 1st line treatment seems to affect survival in mCRPC setting. There is no analysis for this in the current article despite there are many publications in this subject. 

5. Discussion is rather confused mentioning both mCRPC and MHSPC clinical trials without any specific target and no specific conclusions can be raised. 

6. The definition of chemo-naive and chemo-treated mCRPC patients is also rather confusing, since with contemporary treatment paradigm this refers to mHSPC setting. Author should change these terms to upfront ARTA or Chemo-treated mCRPC patients throughout the text. 

7. 

Author Response

This is a multicentre retrospective study on mCRPC patients. Authors have focused their analysis on what they have characterized on previous publication as "High-volume disease"

In the specific article the effect of previous chemotherapy was analyzed. 

There are several problems in this article that should be improved:

  1. Scientific hypothesis for this analysis as stated by the authors is "The lack of a general consensus recommendation for the optimal sequencing of systemic therapeutic agents"  and they have "reported survival outcomes of patients with mCRPC from a longitudinal real-life multicenter series" 

However, what they present is survival outcomes of what considered "High-Volume disease" in chemo-treated vs chemo-naive mCRPC patients. The aim of the analysis should be clearly stated in their introduction. 

We appreciate the reviewer for the comment. The aim of the study was clearly stated in the introduction section and the manuscript was modified as follows: “On this background, we reported survival outcomes of high-volume mCRPC patients treated with upfront-ARTA in a chemo-naïve setting compared to patients treated with chemotherapy as first line in a longitudinal real-life multicenter series”.

  1. What is the role of "High-Volume" disease in the mCRPC setting? There is a general confusion in the article betwen mCRPC and mHSPC setting. The High-volume disease definition was initially introduced in mHSPC setting and its role was confirmed as prognostic biomarker. It also allowed to differentiate treatment pathways between high- and low-volume patients. However, current definition for mCRPC disease lack generalized validation and was not used in consensus statements or guidelines for treatment selection. Other parameters eg. symptomatic disease that was earlier used was not included in the current analysis.

As the reviewer stated, a general consensus and universally accepted definition of high-volume mCRPC have not yet been outlined. We applied a previously published definition of high-volume disease specifically used for mCRPC population. (ref. Ferriero et al., 7 and 11) High tumor burden was defined based on radiologic extent of disease. Symptoms were not considered a reason for defining a patient as high volume.

Therefore, we found that high volume mCRPC patients treated with novel antiandrogens as first-line treatment displayed similar oncological outcomes compared to patients treated with chemotherapy upfront.

  1. In the time period analyzed, chemotherapy and ARTAs were also approved for the mHSPC setting and obviously such treatments affect treatment selection and prognosis in mCRPC. There is no information regarding the disease pathway for these patients and for those that were castration resistant after mHSPC treatment there is no information regarding treatment in the mHSPC setting. 

The aim of this study was to compare the survival outcomes of mCRPC patients with high-volume disease in a chemo-naïve context compared to cases treated with chemotherapy upfront when developed castration resistance, in a longitudinal real-life multicenter series. Patients treated with ARTA or chemotherapy in a castration sensitive setting were excluded. The text was modified accordingly.

Consequently, the selected patients only received ARTA or Docetaxel as first- or second-line treatments specifically during the mCRPC condition; while on hormone sensitive phase, patients were treated only with Lh-RH agonist or antagonist.

  1. Treatment sequencing rather than 1st line treatment seems to affect survival in mCRPC setting. There is no analysis for this in the current article despite there are many publications in this subject. 

As previously mentioned, the objective of this study was to assess the effectiveness of first-line treatments in CRPC patients who were chemo-naïve compared to those who had undergone chemotherapy, within the context of a real-life longitudinal scenario. Specifically, we demonstrated similar survival outcomes in high volume disease regardless chemotherapy or ARTA regimen. While investigating treatment sequencing falls beyond the scope of this manuscript. We already published a paper dealing with management of lines of therapy in CRPC patients (ref. 7)

  1. Discussion is rather confused mentioning both mCRPC and MHSPC clinical trials without any specific target and no specific conclusions can be raised. 

The paper has been written in order to respect the requested length. So that introduction and discussion section widely investigated the current literature on metastatic prostate cancer, including the main registrative trials for both mHSPC and mCRPC scenarios. While we acknowledge that the topic might occasionally appear intricate, it represents the natural history of disease and how long and intense became the cure with the introduction of novel antiandrogen receptors targeted antagonist (ARTA) and other treatment options.

In terms of our specific conclusions, our study highlighted the potential effectiveness of utilizing first-line ARTA in high volume CRPC patients to postpone chemotherapy initiation, thus potentially reducing treatment-related toxicity. However, several limitations weaken the strength and reliability of our conclusions. These include the absence of a widespread consensus on the optimal sequencing for mCRPC patients, the relatively small sample size, the retrospective nature of our study, and the heterogeneity within our study cohort. As a result, drawing definitive and robust conclusions is challenging.

Consequently, our results required confirmation through external validation and prospective studies involving larger and more homogeneous populations.

All these comments were highlighted in discussion section.

  1. The definition of chemo-naive and chemo-treated mCRPC patients is also rather confusing, since with contemporary treatment paradigm this refers to mHSPC setting. Author should change these terms to upfront ARTA or Chemo-treated mCRPC patients throughout the text. 

We thank the reviewer for the suggestion. All sentences were modified with “upfront-ARTA” and “chemo-treated” mCRPC patients. Modifications were highlighted in the main text.

Reviewer 2 Report

In this manuscript Authors aimed to evaluate the effect of upfront chemotherapy versus first-line treatment with androgen receptor targeted agents (ARTA) on progression and survival in high-volume metastatic castration resistant prostate cancer patients using a multi-center collaborative database. 

All contents showed are of interest since this is a still debated topic within the medical community. However, I would share with the Authors some concerns and suggestions to improve the overall quality of their manuscript.

First, I strongly suggest revising the manuscript for English language grammar and syntax, which are important limitations in the process of comprehension of the main text. For example, in multiple instances, Authors relied on very long sentences. Shorter, more concise sentences would greatly improve the readability and the flow of the manuscript.

Second, how did you choose the criteria for defining high-volume disease? How these criteria differed from the CHARTEED and GETUG-AFU15 definitions?

Third, the aim of the study should be better explained in the Introduction. It is ok to state “We reported survival outcomes of patients with metastatic castration resistant prostate cancer from a longitudinal real-life multicenter series”. However, as indicated in the initial sentence of the Discussion the study focused on the comparison between patients receiving chemotherapy or ARTA as first line treatment option. 

Fourth, no results have been shown regarding secondary outcome "PSA response". 

Fifth, another possible limitation is the heterogeneity of the overall population due to the inclusion of either metastatic de novo patients and patients who were not metastatic at initial diagnosis and progressed during the follow-up. 

Sixth, did you consider the possibility to rely on the propensity score matching to better balance the two cohorts reducing the effect of selection bias?

Seventh, please consider citing this article: PMID 35652586. DOI:10.1002/pros.24376

Extensive editing of English language is required.

Author Response

In this manuscript Authors aimed to evaluate the effect of upfront chemotherapy versus first-line treatment with androgen receptor targeted agents (ARTA) on progression and survival in high-volume metastatic castration resistant prostate cancer patients using a multi-center collaborative database. 

All contents showed are of interest since this is a still debated topic within the medical community. However, I would share with the Authors some concerns and suggestions to improve the overall quality of their manuscript.

First, I strongly suggest revising the manuscript for English language grammar and syntax, which are important limitations in the process of comprehension of the main text. For example, in multiple instances, Authors relied on very long sentences. Shorter, more concise sentences would greatly improve the readability and the flow of the manuscript.

The manuscript has been enhanced in accordance with the reviewer's suggestions. Sentences have been shortened, and a comprehensive English review has been conducted by a native English speaker. All modifications have been highlighted within the main text.

Second, how did you choose the criteria for defining high-volume disease? How these criteria differed from the CHARTEED and GETUG-AFU15 definitions?

A general consensus and universally accepted definition of high-volume mCRPC have not yet been outlined. We applied a previously published definition of high-volume disease specifically used for mCRPC population. (ref. Ferriero et al., 9 and 11)

High tumor burden was defined based on radiologic extent of disease, as bulky positive nodes (≥ 5 cm) or more than 6 bone metastases at mCRPC. Symptoms were not considered a reason for defining a patient as high volume.

Our definition appeared more effective in stratifying our cohort based on disease volume, allowing us to easily identify patients who genuinely exhibit high tumor burden.

Third, the aim of the study should be better explained in the Introduction. It is ok to state “We reported survival outcomes of patients with metastatic castration resistant prostate cancer from a longitudinal real-life multicenter series”. However, as indicated in the initial sentence of the Discussion the study focused on the comparison between patients receiving chemotherapy or ARTA as first line treatment option. 

We appreciate the reviewer for their comment. The aim of the study was clearly stated in the introduction section, and the manuscript was modified as follows: “On this background, we reported survival outcomes of high-volume mCRPC patients treated with upfront-ARTA in a chemo-naïve setting compared to patients treated with chemotherapy as first line in a longitudinal real-life multicenter series”.

Fourth, no results have been shown regarding secondary outcome "PSA response". 

Addressing secondary outcomes is beyond the scope of this manuscript, as our primary focus is to present a longitudinal real-life survival outcomes of upfront chemotherapy compared to first-line ARTA in high volume mCRPC patients. Therefore PSA is not considered a reliable prognosticator in this setting of disease, while survival analysis represent the whole role of treatment lines, including radiologic response and toxicity. The sentence in M&M section has been removed.

Fifth, another possible limitation is the heterogeneity of the overall population due to the inclusion of either metastatic de novo patients and patients who were not metastatic at initial diagnosis and progressed during the follow-up. 

The limitations section has been revised accordingly, and the modifications have been highlighted as follows: "Another potential limitation is the heterogeneity of the overall population, due to the inclusion of both metastatic de novo patients and those who were initially non-metastatic but progressed during the follow-up."

Sixth, did you consider the possibility to rely on the propensity score matching to better balance the two cohorts reducing the effect of selection bias?

Even if it is feasible, a PSM analysis would have led to a considerably reduced final population due to the inherent heterogeneity of our study cohorts.

Seventh, please consider citing this article: PMID 35652586. DOI:10.1002/pros.24376

As suggested by the reviewer, the article was cited in the main text and the modification highlighted.

Reviewer 3 Report

Authors should be congratulated for their work. The topic is interesting and intriguing. The right assessment of metastatic prostate cancer patients is the base for the correct and tailored treatment, in order to minimize the exposure to more toxic treatments. The manuscript is well-written and easily readable. However, the images' quality is scarce. The tables should be restructured to fit on a single page. I suggest using only one label (e.g. cN0 or cN1).

A Major revision is required.

Author Response

Authors should be congratulated for their work. The topic is interesting and intriguing. The right assessment of metastatic prostate cancer patients is the base for the correct and tailored treatment, in order to minimize the exposure to more toxic treatments. The manuscript is well-written and easily readable.

We express our gratitude to the reviewer for their willingness to review our manuscript and for taking the time to provide thoughtful comments.

However, the images' quality is scarce. The tables should be restructured to fit on a single page. I suggest using only one label (e.g. cN0 or cN1). A Major revision is required.

We thank the reviewer for the suggestion. The images and tables were improved accordingly and modifications highlighted in the main text.

Reviewer 4 Report

The article provides a comprehensive overview of the context, objectives, patient selection, treatment regimens, follow-up, and statistical analysis methodologies. It appears to be well-structured and informative, focusing on a clinically relevant topic in the field of prostate cancer management. The article aims to contribute to the understanding of treatment outcomes and optimal sequencing in the context of metastatic castration-resistant prostate cancer. Certainly, here's a more critical analysis of the various sections of the article:

Introduction:

The introductory section provides a general overview of the context of metastatic castration-resistant prostate cancer (mCRPC) and the importance of urologists' involvement in the decision-making process. However, it lacks a clear justification of the specific aim of the study and the gaps in the field it intended to address. Additionally, the section does not mention specific references to existing medical literature to support the need for this study.

Materials and Methods:

While some information about the patient population and administered treatments is provided, the section lacks a comprehensive and transparent description of patient selection methods and underlying rationales. It is not mentioned whether patient assignment to various treatments was random or whether variables that could influence the choice were considered. This raises concerns about potential selection biases that could impact the results.

Results:

The results are presented fairly clearly, but there are some significant gaps. The sample size of patients is limited, especially considering the subdivision into subgroups. This makes it difficult to generalize the results to a broader population of mCRPC patients. Additionally, the results are presented in terms of numerical data without clear clinical interpretation, making it difficult to assess the clinical significance of the findings.

Discussion:

The discussion starts well by highlighting the lack of clear guidelines for mCRPC management and the importance of accurate treatment selection. However, the text appears disjointed and lacks clear structure in several parts. Numerous studies are mentioned without a clear connection to the current study. Furthermore, discrepancies between the results of different studies are not adequately explained or contextualized. This compromises the reader's ability to evaluate the implications of the study's results in relation to existing data. The authors should read the following paper and discuss them: PMID: 37446024; PMID: 36294423.

Conclusions:

The conclusions appear to stem from a rather confused discussion and are not convincingly supported by the presented results. Despite considering the results comparable, the lack of clear structure and argumentation in the discussion undermines the robustness of the conclusions. Additionally, the conclusions do not suggest any next steps or future directions for research.

Overall, the article suffers from a lack of clarity in its objective, methodology, and interpretation of results. The discussion appears fragmented and lacks clear logical structure. While the topic addressed is of interest and relevance, deficiencies in the presentation and analysis of data diminish confidence in the validity and applicability of the results.

Author Response

The article provides a comprehensive overview of the context, objectives, patient selection, treatment regimens, follow-up, and statistical analysis methodologies. It appears to be well-structured and informative, focusing on a clinically relevant topic in the field of prostate cancer management. The article aims to contribute to the understanding of treatment outcomes and optimal sequencing in the context of metastatic castration-resistant prostate cancer.

Certainly, here's a more critical analysis of the various sections of the article:

Introduction:

The introductory section provides a general overview of the context of metastatic castration-resistant prostate cancer (mCRPC) and the importance of urologists' involvement in the decision-making process. However, it lacks a clear justification of the specific aim of the study and the gaps in the field it intended to address.

Although the high-volume mCRPC setting continues to warrant upfront chemotherapy, there's a clear need for a unanimous recommendation concerning the most effective sequencing of systemic therapeutic agents. In this context, the utilization of subsequent lines of androgen receptor-targeted agents (ARTA) has been proposed as a viable alternative.

The aim of the study was clearly stated in the introduction section, and the manuscript was modified as follows: “On this background, we reported survival outcomes of high-volume mCRPC patients treated with upfront-ARTA in a chemo-naïve setting compared to patients treated with chemotherapy as first line in a longitudinal real-life multicenter series”.

Additionally, the section does not mention specific references to existing medical literature to support the need for this study.

The challenge within the context of the mCRPC setting lies precisely in the absence of dedicated studies and consequently, references and established recommendations concerning the most effective treatment sequencing, especially in those presenting with high volume disease. The use of chemotherapy upfront often comes with a significant rate of toxicity, and its application is rooted in studies conducted prior to the era of ARTA. As a result, a unanimous consensus recommendation that outlines the optimal sequencing of systemic therapeutic agents remains a pressing need.

The text has been modified in order to highlight the issues and support the need of our study.

Materials and Methods:

While some information about the patient population and administered treatments is provided, the section lacks a comprehensive and transparent description of patient selection methods and underlying rationales. It is not mentioned whether patient assignment to various treatments was random or whether variables that could influence the choice were considered. This raises concerns about potential selection biases that could impact the results.

In this study, we present data derived from an analysis of a longitudinal real-life cohort of mCRPC patients across multiple centers. The retrospective analysis could raise concerns regarding potential selection biases and may influence the outcomes. Considering this context, our study aims to offer a portrayal of a real-world scenario, thereby producing results that are more broadly generalizable. These results can be applied to a larger population of high-volume mCRPC patients, regardless their baseline characteristics and demographic profiles. Specifically, the decision to assign patients to either ARTA treatment or upfront chemotherapy was influenced by factors such as comorbidities, genomics, and variations in treatment preferences between urologists and oncologists, or drug availability.

This issue has been mentioned as a limitation of the study.

Results:

The results are presented fairly clearly, but there are some significant gaps. The sample size of patients is limited, especially considering the subdivision into subgroups. This makes it difficult to generalize the results to a broader population of mCRPC patients. Additionally, the results are presented in terms of numerical data without clear clinical interpretation, making it difficult to assess the clinical significance of the findings.

The issue regarding the sample size was already stated in the limitations section (“The small unbalanced sample size and the lack of control over potential confounding factors for treatment response may impact on our results”). Nonetheless, the longitudinal real-life experience enhanced our findings, displaying a higher level of generalizability compared to those derived from a prospective RCT with strict inclusion and exclusion criteria. External validation is imperative to corroborate the outcomes of our study and to gain the most appropriate patient selection process for a specific upfront treatment regimen.

In conclusion, we showed comparable survival outcomes, regardless tumor volume and the choice between a chemotherapy or non-chemotherapy regimen. This observation strengthen the use of ARTA as first treatment line even in hjgh volume mCRPC patients.

Discussion:

The discussion starts well by highlighting the lack of clear guidelines for mCRPC management and the importance of accurate treatment selection. However, the text appears disjointed and lacks clear structure in several parts. Numerous studies are mentioned without a clear connection to the current study. Furthermore, discrepancies between the results of different studies are not adequately explained or contextualized. This compromises the reader's ability to evaluate the implications of the study's results in relation to existing data. The authors should read the following paper and discuss them: PMID: 37446024; PMID: 36294423.

We thank the reviewer for the suggestion. The paper has been written in order to respect the requested lenght. So that introduction and discussion sections widely investigated the current literature on metastatic prostate cancer, including the main registrative trials for both mHSPC and mCRPC scenarios. While we acknowledge that the topic might occasionally appear intricate, it represents the natural history of disease and how long and intense became the cure with the introduction of novel antiandrogen receptors targeted antagonist (ARTA) and other treatment options.

Nevertheless, the discussion was revised and improved.

We carefully read the suggested interesting papers; however, we respectfully found hard to connect them with the topic of the present article. It could be a good idea for a future paper on the biological activities of advanced prostate cancer.

Conclusions:

The conclusions appear to stem from a rather confused discussion and are not convincingly supported by the presented results. Despite considering the results comparable, the lack of clear structure and argumentation in the discussion undermines the robustness of the conclusions. Additionally, the conclusions do not suggest any next steps or future directions for research.

Overall, the article suffers from a lack of clarity in its objective, methodology, and interpretation of results. The discussion appears fragmented and lacks clear logical structure. While the topic addressed is of interest and relevance, deficiencies in the presentation and analysis of data diminish confidence in the validity and applicability of the results.

As requested, the discussion was revised and improved, suggested studies were cited in order to make conclusions clearer.

Our study highlighted a noteworthy observation: in carefully chosen patients, mCRPC treatment with first-line ARTA could potentially be an effective strategy to delay a more toxic chemotherapy.

However, the absence of a widespread agreement on the optimal sequencing for mCRPC patients, the limited sample size, the retrospective nature of analysis and the heterogenity of the patient cohort, weaken our findings.

Consequently, it becomes imperative to confirm our results through external validation and prospective studies with a larger and more homogenous population.

Round 2

Reviewer 2 Report

Authors properly replied all my previous contents. 

The manuscript is suitable for publication in the current form.

Reviewer 3 Report

Authors should be congratulated for their work. They improved the quality of the manuscript following the suggestions of the reviewers. The manuscript is suitable for publication in its current form

Reviewer 4 Report

the paper it is with of publication in the current form